# A standardized metric to enhance clinical trial design and outcome interpretation in type 1 diabetes

Alyssa Ylescupidez [1], Henry T. Bahnson[1], Colin O'Rourke[1], Sandra Lord[1], Cate Speake [1,2] ✉ & Carla J. Greenbaum [1,2] ✉

The use of a standardized outcome metric enhances clinical trial interpretation and cross-trial comparison. If a disease course is predictable, comparing modeled predictions with outcome data affords the precision and confidence needed to accelerate precision medicine. We demonstrate this approach in type 1 diabetes (T1D) trials aiming to preserve endogenous insulin secretion measured by C-peptide. C-peptide is predictable given an individual's age and baseline value; quantitative response (QR) adjusts for these variables and represents the difference between the observed and predicted outcome. Validated across 13 trials, the QR metric reduces each trial's variance and increases statistical power. As smaller studies are especially subject to random sampling variability, using QR as the outcome introduces alternative interpretations of previous clinical trial results. QR can provide model-based estimates that quantify whether individuals or groups did better or worse than expected. QR also provides a purer metric to associate with biomarker measurements. Using data from more than 1300 participants, we demonstrate the value of QR in advancing disease-modifying therapy in T1D. QR applies to any disease where outcome is predictable by pre-specified baseline covariates, rendering it useful for defining responders to therapy, comparing therapeutic efficacy, and understanding causal pathways in disease.

Ideally, a metric for studying disease should be clinically and scientifically meaningful, objective, predictable, and able to be standardized across individuals and cohorts. When applied in the context of clinical trials for any disease, such a standardized metric may enable acceleration of trials through increased statistical power and aid in interpretation of clinical trial data by regulators, clinicians, investigators, translational scientists, and study participants.

If a disease course or outcome is predictable using baseline factors, analysis should adjust for these factors as long as they are specified in advance[1]. This approach is advantageous over traditional unadjusted analysis, which essentially compares the average of the group of treated individuals to the average of the group of control individuals. Baseline covariate adjustment improves precision for estimating treatment effects of drugs and biological products, and "covariate adjustment leads to efficiency gains when the covariates are prognostic for the outcome of interest in the trial" and are pre-specified in the statistical analysis plan[2]. In unadjusted analysis, results may be more strongly impacted by chance covariate imbalances at baseline, especially when there is evidence that the covariate is associated with the outcome, obscuring the effect of treatment. Despite the known benefits of baseline-adjusted analyses, reviews have found that only 24–34% of trials use covariate adjustment for the primary analysis[3].

A standardized quantitative response (QR) metric that adjusts for baseline covariates can be developed for any reproducible outcome

[1]Center for Interventional Immunology, Benaroya Research Institute at Virginia Mason, Seattle, WA, USA. [2]These authors contributed equally: Cate Speake, Carla J. Greenbaum. ✉e-mail: cspeake@benaroyaresearch.org; cjgreen@benaroyaresearch.org

measure for which the natural history is known and predictable. This is the case for trials of disease-modifying therapy (DMT) in type 1 diabetes (T1D) aiming to preserve endogenous pancreatic beta cell function, as there is a wealth of natural history data on the loss of insulin secretion over time measured by the C-peptide response to a mixed meal tolerance test (MMTT)[4–18]. Moreover, though it is noteworthy that a therapy to delay onset of clinically apparent disease was recently approved for clinical use[19], there are still no DMTs that preserve endogenous insulin secretion in individuals recently diagnosed with T1D and only one in the prevention setting. Preservation of insulin secretion after diagnosis is associated with improved clinical outcomes[20–24]. To date, there have been a few dozen randomized controlled trials (RCT) of immune therapy in recently (<3 months) diagnosed T1D and almost all are phase 2 trials led by academic investigators. Further, the time to conduct such studies is long, given that trials are challenging to enroll and study endpoints are at least 1 year from randomization. While C-peptide response to a MMTT is accepted as the appropriate measure of endogenous insulin secretion[20,25], regulatory ambiguity for potential indications exists since there is no established clinical therapeutic threshold of C-peptide that definitively qualifies therapies or interventions as successes or failures[26]. In addition, by design, published trials of immune therapy express the average effect of therapy on the randomized cohort and, though multiple definitions have been proposed, there is no accepted standardized criteria to define a responder to therapy. Together, these issues create limitations in understanding mechanisms of disease and response to therapy, hindering the ability to develop a precision medicine approach to DMT in T1D and other immune-associated diseases.

The QR, originally developed by Bundy and Krischer[27], leverages the well-known statistical property that model adjustment with prognostic baseline covariates increases precision and confidence by way of controlling for outcome heterogeneity[3]. Bundy and Krischer used data from five trials[6–8,11,12] in similar populations to develop an analysis of covariance (ANCOVA) model to predict the C-peptide area under the curve (AUC) mean value by adjusting for baseline C-peptide AUC mean and age. The resulting QR metric is a standardized measure of the difference between an individual's observed and predicted C-peptide AUC mean one year after study entry. Values above zero indicate a better-than-expected outcome and values below zero indicate a worse-than-expected outcome[27].

Using data from 13 RCTs testing 14 different therapies aiming to preserve beta cell function in those with T1D, we demonstrate how the QR metric increases the precision and confidence of clinical trial results, thus enhancing interpretation of these studies while suggesting new concepts for future trial designs. We found that the QR metric reduced variance and standardized C-peptide outcomes across trials, leading to re-evaluation and new interpretations of both clinical and mechanistic results. In addition, we illustrate how the QR metric may be useful for design of future trials. Together, these findings represent a significant step towards precision medicine.

## Results

### The QR metric reduces variance and standardizes C-peptide outcomes across trials

We first validated the published QR metric using data from 13 studies: five TrialNet RCTs used in the development of QR (referred to as the development cohort)[6–8,11,12], and eight additional RCTs (referred to as the validation cohort)[9,10,13–18]. To further evaluate the published model, a new ANCOVA model was also fit to the placebo participants in only the validation cohort. The predicted 1-year C-peptide values from this new model were then used to compute a new QR metric. These newly computed QR values were highly correlated with the QR values from the published model (r = 0.996 in validation cohort, 0.992 in development cohort), and it was thus determined that the published QR

model is applicable to all 13 RCTs (Supplemental Fig. 1A and B). Table 1 lists the key characteristics of each of the 13 trials, including number of subjects, median age, and baseline C-peptide AUC mean. As expected, the mean QR in the development cohort centers around zero, this was also seen in the validation cohort. We found that the mean QR matches closely between the development and validation cohorts, and the distribution of QR is similar in both cohorts (Fig. 1; p = 0.43, two-sample t-test [t = 0.8, DF = 346]; p = 0.62, Kolmogorov-Smirnov two-sample test [KS = 0.04]).

We next updated the model to include all available data from all 13 studies. This revised ANCOVA model predicted 1-year C-peptide AUC mean values very similar to those predicted from the published QR model (Supplemental Fig. 1C, D). Moreover, the predicted values between the original and revised models were strongly associated (R² = 0.998; Supplemental Fig. 1E); thus, it was determined that the original QR model applied well to all 13 studies and all results hereafter use the original QR model. In addition, the model is robust; predictions were counterintuitive for only eight of the 1306 individuals studied. For these eight older individuals with low C-peptide, the model predicted a very minor increase in C-peptide over time (Supplemental Fig. 2).

The use of the QR metric both reduced the variance and standardized the mean of the C-peptide outcome within each trial. Among the placebo-treated individuals in the 13 trials, there were noteworthy variations with respect to both age and baseline C-peptide between studies (Fig. 2a). The 1-year C-peptide AUC mean value varied (mean range 0.36–0.69 nmol/L) between trials, and within each trial demonstrated wide heterogeneity (Fig. 2b). In contrast, by accounting for baseline C-peptide and age, the mean QR value of placebo-treated individuals centered around zero for each trial (mean range −0.07 to 0.08) (Fig. 2b). For 12 of these 13 trials, the mean QR value was not statistically different from zero; the only exception being the TrialNet ATG/GCSF trial (mean of −0.072, p = 0.031; 30 placebo participants). For each individual trial, the standard deviation of the QR was lower than the standard deviation using the C-peptide AUC mean (Fig. 2b, annotated in blue).

We next investigated whether the QR metric would increase statistical power since covariate adjustment in randomized trials leads to greater power and better control of type I and type II error[3]. We chose to examine this in the Immune Tolerance Network (ITN) AbATE trial of teplizumab, which had a positive outcome in demonstrating efficacy of teplizumab in preserving beta cell function in recently diagnosed individuals relative to controls[13]. In the original analysis, the primary outcome used the difference in 4-h C-peptide AUC mean between baseline and 2 years, adjusted for baseline C-peptide, with a p-value of 0.002 for the difference between treatment groups. In our re-analysis, we used the 2-h C-peptide AUC mean at 1 year and found that the difference in C-peptide AUC mean in control (0.364 nmol/L) compared to teplizumab-treated (0.647 nmol/L) at 1 year was statistically significant with a p-value of 0.009 (t = 2.7, DF = 46.4) using a two-sample t-test assuming unequal group variances (Fig. 3a). When using the change from baseline C-peptide AUC mean as in the published trial report, the effect size and precision increased with a mean change from baseline of −0.321 nmol/L in the control group and −0.086 nmol/L in the teplizumab-treated group (p = 0.0002, t = 4.0, DF = 57.0) (Fig. 3b). Controlling for both baseline C-peptide and age by using QR as the outcome further increased the statistical significance of the result (−0.015 nmol/L control vs 0.141 nmol/L teplizumab-treated, p < 0.0001, t = 4.3, DF = 47.3) (Fig. 3c).

We next assessed whether the QR approach could be utilized to predict trial outcomes beyond 1 year, using varying baseline reference points and outcome timepoints. The original QR model used baseline values from individuals within three months of diagnosis to predict the outcome at 12 months after treatment initiation and demonstrated an R² value of 53%. Predictions further in the future are more challenging, thus it is not surprising that the R² drops when this same baseline is

**Table 1 | Characteristics of trials used in analysis**

| Trial | Start Year | N Treatment Arm(s) | N Control Arm | Age, median (range) | Baseline C-peptide AUC Mean (nmol/L), mean (SD) | Primary outcome/ time point | Primary Study Result |
|---|---|---|---|---|---|---|---|
| Diamyd Therapeutics AB: GAD-Alum Phase 2 | 2005 | 35 | 35 | 13.9 (10.1, 18.4) | 0.66 (0.37) | Fasting C-peptide at 15 months | Negative |
| Diamyd Therapeutics AB: GAD-Alum Phase 3 | 2008 | 107 (2 doses), 109 (4 doses) | 111 | 13.2 (10, 19.3) | 0.66 (0.32) | 2-h MMTT AUC C-peptide at 15 months | Negative |
| Immune Tolerance Network (ITN): Anti-thymocyte globulin (ATG) | 2007 | 38 | 20 | 17.5 (12, 35) | 0.88 (0.42) | 2-h MMTT AUC C-peptide at 1 year | Negative |
| Immune Tolerance Network (ITN): Alefacept | 2011 | 33 | 16 | 18 (12, 34) | 0.78 (0.38) | 2-h MMTT AUC C-peptide at 1 year (secondary 4-h MMTT AUC C-peptide at 1 year) | Negative primary; Positive secondary |
| Immune Tolerance Network (ITN): Teplizumab | 2005 | 52 | 25 | 12 (8.3, 29.6) | 0.7 (0.3) | 4-h MMTT AUC C-peptide at 2 years | Positive |
| Immune Tolerance Network (ITN): Tocilizumab | 2015 | 88 | 47 | 14 (6, 45) | 0.76 (0.43) | 2-h MMTT AUC C-peptide at 1 year | Negative |
| JDRF: Imatinib | 2014 | 43 | 21 | 24.5 (18.3, 45) | 0.84 (0.4) | 2-h MMTT AUC C-peptide at 1 year | Positive |
| TrialNet: Abatacept | 2008 | 76 | 35 | 12.9 (6.4, 36.8) | 0.75 (0.38) | 2-h MMTT AUC C-peptide at 2 years | Positive |
| TrialNet: Canakinumab | 2010 | 48 | 22 | 11 (6, 31.9) | 0.64 (0.33) | 2-h MMTT AUC C-peptide at 1 year | Negative |
| TrialNet: GAD-Alum | 2009 | 49 (2 doses), 48 (3 doses) | 48 | 15.1 (3.5, 45.7) | 0.73 (0.32) | 2-h MMTT AUC C-peptide at 1 year | Negative |
| TrialNet: Low Dose ATG/GCSF | 2014 | 29 (ATG only), 28 (ATG/GCSF) | 31 | 16 (12, 42.5) | 0.89 (0.44) | 2-h MMTT AUC C-peptide at 1 year | Positive |
| TrialNet: MMF/DZB | 2004 | 31 (MMF only), 41 (MMF/DZB) | 42 | 14.9 (8.7, 46.1) | 0.69 (0.32) | 2-h MMTT AUC C-peptide at 1 year | Negative |
| TrialNet: Rituximab | 2005 | 55 | 30 | 16 (8.3, 40.4) | 0.75 (0.39) | 2-h MMTT AUC C-peptide at 1 year | Positive |

used to predict a 24-month outcome (Fig. 4). However, if individuals are enrolled at 6, 12, or 18 months from diagnosis, the $R^2$ value for an outcome at two years is high (74%, 85%, 87%, respectively), suggesting that QR can be used as an outcome measure for trials enrolling individuals further from diagnosis. Since most trials evaluating immune therapies have been conducted in individuals within 3 months of diagnosis, this suggests the possibility of trials designed to test therapies with different enrollment windows.

**Applying the QR metric to previous published clinical trials can change the interpretation of both clinical and mechanistic results**

Since the QR metric incorporates historical data from many placebo/control individuals, it minimizes the random sampling variability often present in individual studies with small sample sizes. We determined whether using the QR metric would alter the interpretation of clinical trial results, compared to the originally published reports. In Fig. 5, we show the mean QR (±95% confidence intervals) for the active treatments (Fig. 5a) and placebo/control arms (Fig. 5b), as well as the treatment effect expressed as the difference in QR between arms (Fig. 5c) for 13 published trials.

Expressing the overall treatment effect and results of each arm using QR altered the interpretation of some of the published results. For example, the primary outcome of the alefacept trial was the 2-h C-peptide AUC at 1 year, and the difference between treatment arms did not reach statistical significance in the original analysis[14]. However, applying the QR metric to the alefacept trial dataset demonstrated a large effect in the active treatment group (Fig. 5a), strongly suggesting that alefacept, or drugs working in the same pathway, are worth pursuing in future trials.

For the canakinumab trial, assessing the overall trial result by the difference in treatment arms using the QR metric finds no treatment effect (Fig. 5c), consistent with the published outcome[8]. Yet, the QR of the active arm of the canakinumab trial suggests a positive effect of this therapy on C-peptide (Fig. 5a). Moreover, the QR of the placebo arm allowed us to further interpret this result, revealing that the lack of statistical significance between the groups may be driven by higher-

than-expected C-peptide response in the 22 individuals in the placebo arm of the study (Fig. 5b).

Lastly, applying the QR metric to the two studies testing anti-thymocyte globulin (ATG) also suggests a different interpretation than originally published. The TrialNet ATG/GCSF trial reported a positive outcome[9] using lower-dose therapy than the ITN ATG study, which did not meet its primary outcome[28], leading to the interpretation that dose level was the key variable in the effectiveness of the drug. However, the mean QR in the treated participants was notably similar between studies: mean (95% CI) QR was 0.08 (0.02–0.13) in the higher-dose ITN ATG trial, 0.09 (0.03–0.15) in the lower-dose TrialNet ATG/GCSF trial. This suggests that the reported difference in treatment effect between the studies was driven by the placebo participants: those in the TrialNet ATG/GCSF study had a fairly low mean QR (−0.07) while those in the ITN ATG study had a higher mean QR (0.05). Of note, across the 13 studies, only the TrialNet ATG/GCSF control arm was statistically significantly different from zero. This analysis suggests that rather than drug dosing, the differing behavior of the placebo groups could alternatively explain differences in reported trial outcomes. This further demonstrates how random sampling variability in smaller studies can complicate interpretation of RCT results.

We also looked at the applicability of the QR model to timepoints prior to 1 year. If the mean QR ± 95% CI were used as an outcome at 6 months, we found no treatments that showed a false positive result—that is, all trials positive at 6 months were still positive at 12 months. However, the converse is not true; we show that three therapies positive at 12 months could have been missed using this method (Fig. 6).

Next, we investigated whether using the QR metric would impact the results of immune marker studies aiming to explore mechanisms of response to therapy. Using C-peptide and mechanistic results obtained from the ITN AbATE trial of teplizumab, we confirmed previous reports of the positive association between the frequency of treatment-induced KLRG1 + TIGIT + CD8 + T cells, a known signature of T cell exhaustion, and C-peptide outcome (Fig. 7a)[29]. Adjusting for baseline C-peptide and age by using the QR metric showed a weaker association between treatment-induced

KLRG1 + TIGIT + CD8 + T cells and outcome (Fig. 7b). This observation is likely accounted for in part by an association between treatment-induced exhausted T cells and age (Fig. 7c). As previously noted, age is one of two key variables in the QR metric; age is also known to be important in defining setpoints and responsiveness for many immune cell populations (recently reviewed in refs. 30–33). This analysis implies that therapy-induced exhaustion of T cells unveils mechanistic insights about age itself, which may or may not be causally related to a particular therapy, but is important to our understanding of the role age plays in disease progression and response to therapy. Supplemental Fig. 3 graphically illustrates why QR is a more powerful metric for identification of a biomarker that is causally related to therapy.

### The QR metric better quantifies responders to therapy

As in other diseases, not all individuals recently diagnosed with T1D will respond to a given therapy. Although continuous measurements are preferred to minimize loss of statistical power, historically, analyses of clinical trial results across many diseases frequently stratify treated individuals as responders and non-responders to therapy. In T1D trials, varying definitions of response to therapy using C-peptide have been used[13,34,35]. Reasoning that previously published responder definitions may be associated with baseline variables, we investigated whether the standardized QR metric could better quantify responders to therapy.

We first explored the relationship between baseline C-peptide, age, and the previously published categories of a C-peptide responder/non-responder[13,34,35]. As shown in Fig. 8, among placebo/control participants, the probability of meeting each of the four responder definitions is strongly associated with age (Fig. 8 panels a [Likelihood Ratio (LR) = 11.5, $p = 0.0007$], b [LR = 11.3, $p = 0.0008$], c [LR = 19.8, $p < 0.0001$], d [LR = 11.5, $p = 0.0007$]); two of these definitions are also associated with baseline C-peptide (Fig. 8a [LR = 5.1, $p = 0.02$] and c [LR = 5.4, p = 0.02]). In contrast, the probability of being a responder using the QR-based definition of above or below zero, is, as expected, not associated with either age (LR = 1.8, $p = 0.18$) or baseline C-peptide (LR = 0.2, $p = 0.62$) (Fig. 8e).

To further illustrate the consequences of not accounting for baseline variables in classifying responders, we benchmarked the probability of being a responder for each definition. The average age (16.4 years) and average baseline C-peptide AUC mean for treated individuals across all 13 studies was determined, yielding a QR value of 0.039. However, the probability that an individual with these characteristics is defined as a treatment responder varies widely using different responder definitions (Fig. 8b, c). Most importantly, the probability of being a treatment responder is strongly associated with age, baseline C-peptide, or both of these metrics for all non-QR definitions. Since the QR metric adjusts for age and baseline C-peptide, the probability of being a responder is not conditional on these factors, as can also be seen from the annotated $p$-values (Fig. 8e).

Given that the probability of being a treatment responder is not conditional on age and baseline C-peptide, we asked how the QR metric could be used to select a threshold for classifying responders and non-responders to therapy. In selecting a threshold for a continuous measure such as QR, it is useful to understand the variability or confidence intervals around a given QR value, reflecting the probability that a given QR value represents a true treatment responder. Here, we observed that the distribution of the QR scores of all placebo/control individuals is symmetrical (Fig. 9a), leading to quantile statistics whereby the QR value can be assigned to a percentile (e.g., a QR of 0.10 corresponds to the 75th percentile of the distribution). Figure 9a also illustrates that while there is a symmetrical distribution around the mean, heterogeneity is also apparent; a placebo-treated individual may have a QR value ranging from −0.58 to 0.45. Similarly, though the mean

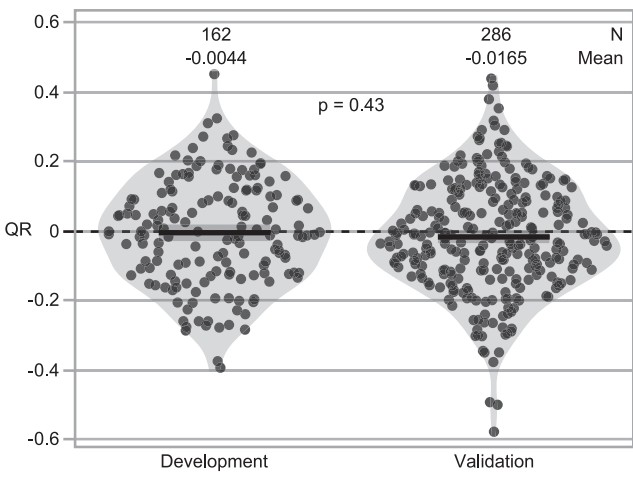

| QR Development vs Validation | Trial | N |
|---|---|---|
| Development | TrialNet: Abatacept | 30 |
| Development | TrialNet: Canakinumab | 19 |
| Development | TrialNet: GAD-Alum | 46 |
| Development | TrialNet: MMF/DZB | 38 |
| Development | TrialNet: Rituximab | 29 |
| Validation | Diamyd Therapeutics AB: GAD-Alum Phase 2 | 34 |
| Validation | Diamyd Therapeutics AB: GAD-Alum Phase 3 | 109 |
| Validation | ITN: Alefacept | 12 |
| Validation | ITN: ATG | 17 |
| Validation | ITN: Teplizumab | 19 |
| Validation | ITN: Tocilizumab | 44 |
| Validation | JDRF: Imatinib | 21 |
| Validation | TrialNet: Low Dose  ATG/GCSF | 30 |

**Fig. 1 | Validation of QR model: comparison of development and validation cohorts.** Placebo/control participants used for QR development ($n = 5$ studies) and validation ($n = 8$ studies). The published QR model ($QR_i = \ln (Cp_{1year,i} + 1) - 0.812 \cdot \ln (Cp_{0,i} + 1) - 0.00638 \cdot Age_i + 0.191$) from the development studies ($n = 5$) was applied to the validation studies ($n = 8$). Since the QR model was derived from the development cohort, it was expected that the QR distribution would be centered at zero. Thus, to assess how well the model fit the validation cohort, we compared the mean and distribution to that of the development cohort. No significant difference was observed between the development and validation cohorts when comparing group means using a two-tailed $t$-test ($t = 0.8$, DF = 346, $p = 0.43$); both cohorts have similar distributions as evaluated by a Kolmogorov-Smirnov test (KS = 0.04, $p = 0.62$). Mean and SD are indicated by black lines and dark shaded region; violin plots visualize data distribution.

QR value among all individuals in the treatment arms of the positive studies is above zero, there is also a wide range of values in each treated group (Fig. 9b), many of which overlap with the distribution of placebo-treated individuals.

The overlapping QR values between treatment and placebo groups demonstrate that using a particular QR cutoff will not necessarily distinguish individuals who received an efficacious therapy from placebo individuals. We reasoned that these distributions can be used to understand the probability that a specific QR value is associated with a successful treatment (Fig. 9c). As shown in Fig. 9c, the probability of identifying a treatment responder or non-responder increases at the extremes of the distribution. For example, the probability that an individual with QR = 0.4 received an effective therapy exceeds 80% (Fig. 9a). Conversely, the probability that an individual with QR = −0.4 received an effective therapy is only 15%: at this threshold, individuals are more likely to be placebo participants, and thus we can infer that a treated

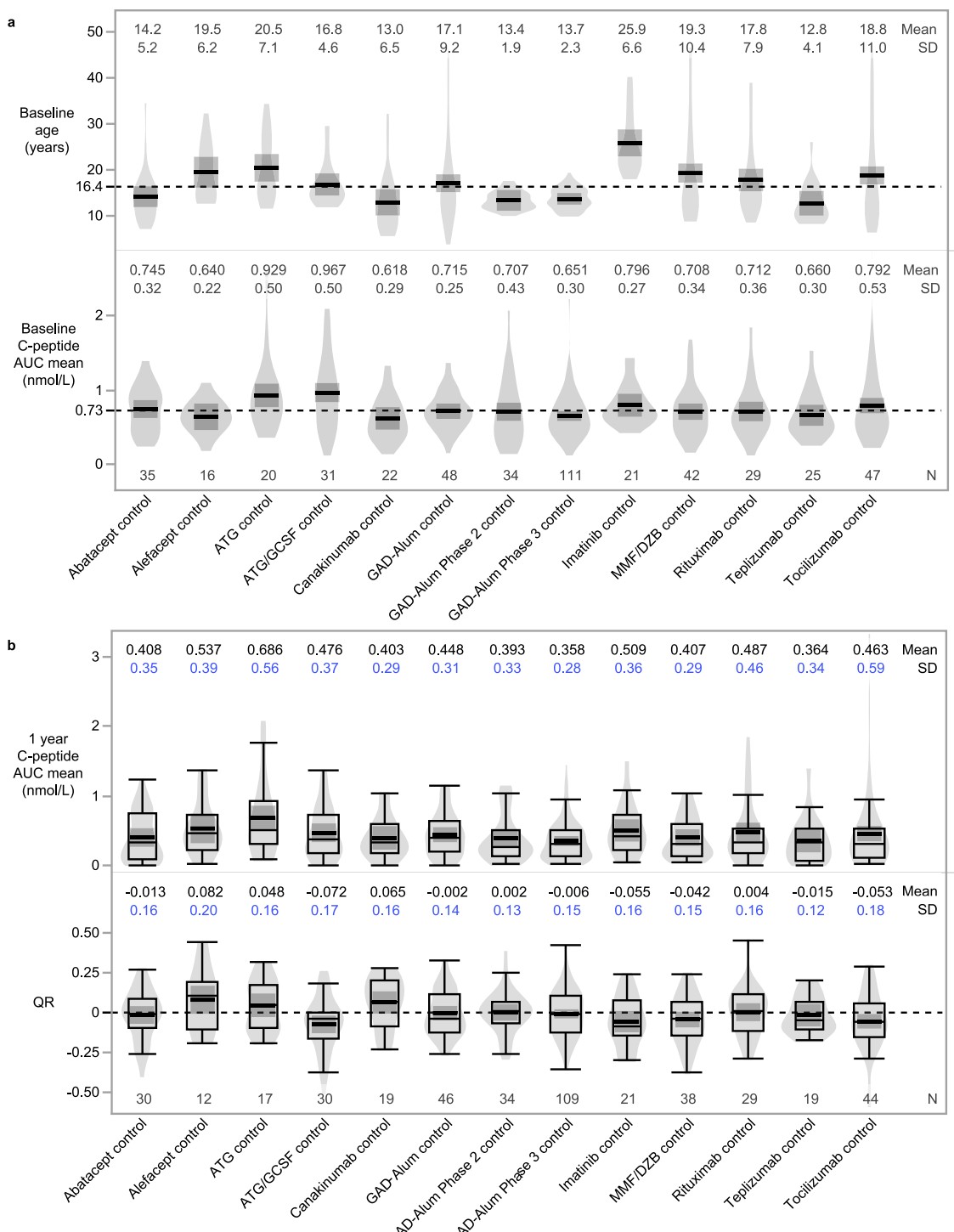

**Fig. 2 | Use of model adjusting for baseline C-peptide and age reduces variance in outcome measure among placebo/control individuals.** Distribution across trials of baseline variables (**a**) of age and C-peptide AUC mean, and outcome measures (**b**) of 1 year C-peptide AUC mean and 1-year QR. Use of QR instead of 1-year C-peptide AUC mean reduces standard deviation (SD; blue) for each trial. Mean and SD are indicated by black lines and dark shaded region. Dashed reference lines in panel A indicate average value in combined placebo participants: 16.4 years

and baseline C-peptide of 0.73 nmol/L. Dashed reference line in panel B indicates a QR of zero to more easily decipher between positive (above reference line) and negative (below reference line) outcomes. Box plots show the distribution of data, excluding outliers; whiskers indicate minimum and maximum, box indicates 1st and 3rd quartiles (interquartile range), and median is represented by thin horizontal line within box. Violin plots visualize data distribution.

individual with this QR value was likely a non-responder to therapy. Choosing less extreme cut points introduces greater uncertainty. For example, selecting a threshold of treatment response of QR = 0.2 would yield only a 65% probability that this individual received an effective therapy.

## Discussion
Despite decades of clinical trials of DMT in individuals recently diagnosed with T1D, there are no drugs currently in clinical practice. Here, we have demonstrated that the QR metric may address many challenges to the field, facilitating the identification of potentially effective

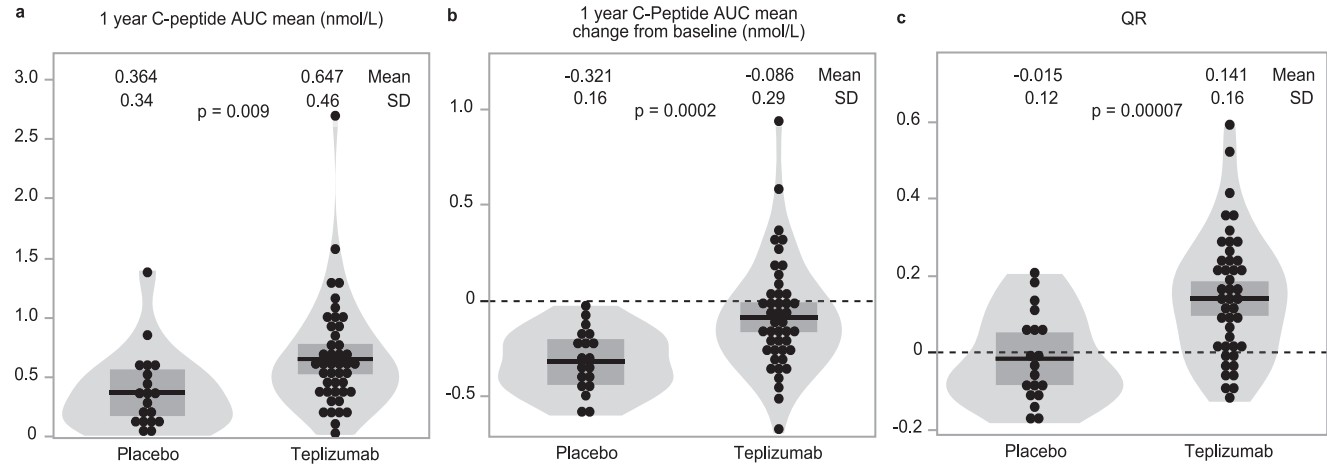

Fig. 3 | More statistical precision with use of QR in teplizumab trial. Outcome data from teplizumab-treated (n = 44) vs control arm (n = 19) using outcome defined as (a) one-year C-peptide AUC mean (nmol/L) difference, t = 2.7, DF = 46.4, p = 0.009; b change from baseline C-peptide AUC mean (nmol/L) difference, t = 4.0

DF = 57.0, p = 0.0002; or (c) one-year QR difference, t = 4.3, DF = 47.3, p = 0.00007. Mean and SD are indicated by black lines and dark shaded region. P-values determined using two-tailed t-test. Dashed lines in panel (b) and (c) are reference lines for unchanged C-peptide (panel b) or QR of 0 (panel c).

therapies. Importantly, standardization of outcomes enables a uniform method of analysis across trials, and thus a manner for comparing therapies and identifying responders to therapies through a consistent responder definition.

We applied the QR metric, which adjusts for baseline age and C-peptide AUC mean, to data from 13 clinical trials of DMT in individuals with recently diagnosed T1D. Since these 13 trials occurred over a 10-year period, included individuals from 3 to 46 years of age, were conducted at multiple locations, and included data from both academic trials and a phase 3 industry-sponsored study, the strength of the model is sufficiently robust to be considered for regulatory purposes.

Whereas traditional unadjusted analysis may be impacted by chance imbalances in covariates at baseline (especially those known to be associated with outcome), baseline-adjusted analysis can lead to individual-specific (conditional) estimates which conceptually match individuals in the intervention group and control group who are similar with respect to the adjusted variables. Using pre-specified variables, baseline-adjusted analysis increases statistical power, allowing for robust comparisons between studies.

In T1D, more than half of the heterogeneity in the natural history of disease can be explained by age and baseline C-peptide. While several T1D trials used ANCOVA models adjusted for baseline metrics, this was inconsistent between studies. Computing a QR further utilizes those ANCOVA predictions[27,36] to determine a standardized score, which enables cross-trial analysis. Analyzing treatment effects in terms of QR also allows for evaluation of treatment groups in a standardized manner, with comparisons to a large number of controls. This approach is particularly powerful in T1D since the baseline covariates are established predictors of the outcome.

Using data from the ITN trial of teplizumab, we demonstrate that using the QR metric reduces the variance of the outcome, resulting in increased power and the potential for reducing sample size. However, it is not clear that reducing the sample size for a phase 2 RCT is the optimal approach to select promising therapies, or to identify responders for T1D or other diseases. Placebo-controlled randomized trials have clear advantages, as randomization can account for potential differences in variables that are known to impact outcome. However, when small sample sizes are used, random sampling variation can significantly impact inferences about trial outcomes. In the case of trials of DMT aiming to preserve C-peptide, the known factors are baseline C-peptide and age; QR adjusts for these factors.

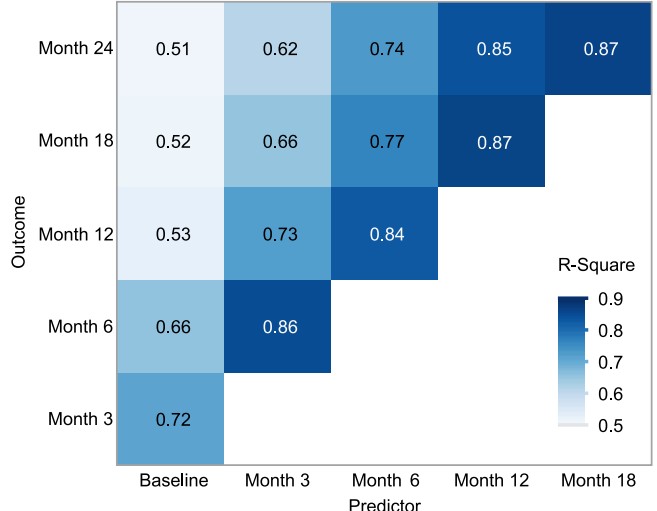

Fig. 4 | Relationship between observed and expected outcome according to time of baseline and time of outcome allows for different trial durations. $R^2$ using baseline C-peptide (within 3 months of diagnosis) and month 12 outcome from original QR model is 53%. Additional ANCOVA models were developed, using different baseline reference points and prediction horizons ranging from 3 months to 2 years. Using C-peptide and age at different time points from diagnosis can also predict outcome; the $R^2$ decreases the longer from the initial measurement. Using predictors at later time points has strong association with outcomes illustrated up to 24 months. Due to differences in visit schedules between studies, n varies at different timepoints: n = 414 at 3 months, n = 461 at 6 months, n = 280 at 9 months, n = 448 at 1 year, n = 277 at 18 months, n = 259 at 24 months. Darker blue coloration indicates stronger correlation.

Using data from almost 500 control/placebo individuals in the 13 trials studied, we show that untreated individuals' outcomes are reliably predicted. Utilizing QR as an outcome measure implies comparison of a treatment arm to this large number of historical controls, and could allow trialists to consider studies without a contemporaneous control group in early phase trials, for example when use of a placebo is not feasible. Single- or multiple-active arm phase 2 trials are likely to conserve resources by eliminating or minimizing placebo participants while accelerating recruitment (as more participants agree to trials without a placebo arm). Of course, caution is always needed in interpretation of studies without contemporaneous controls. However, we

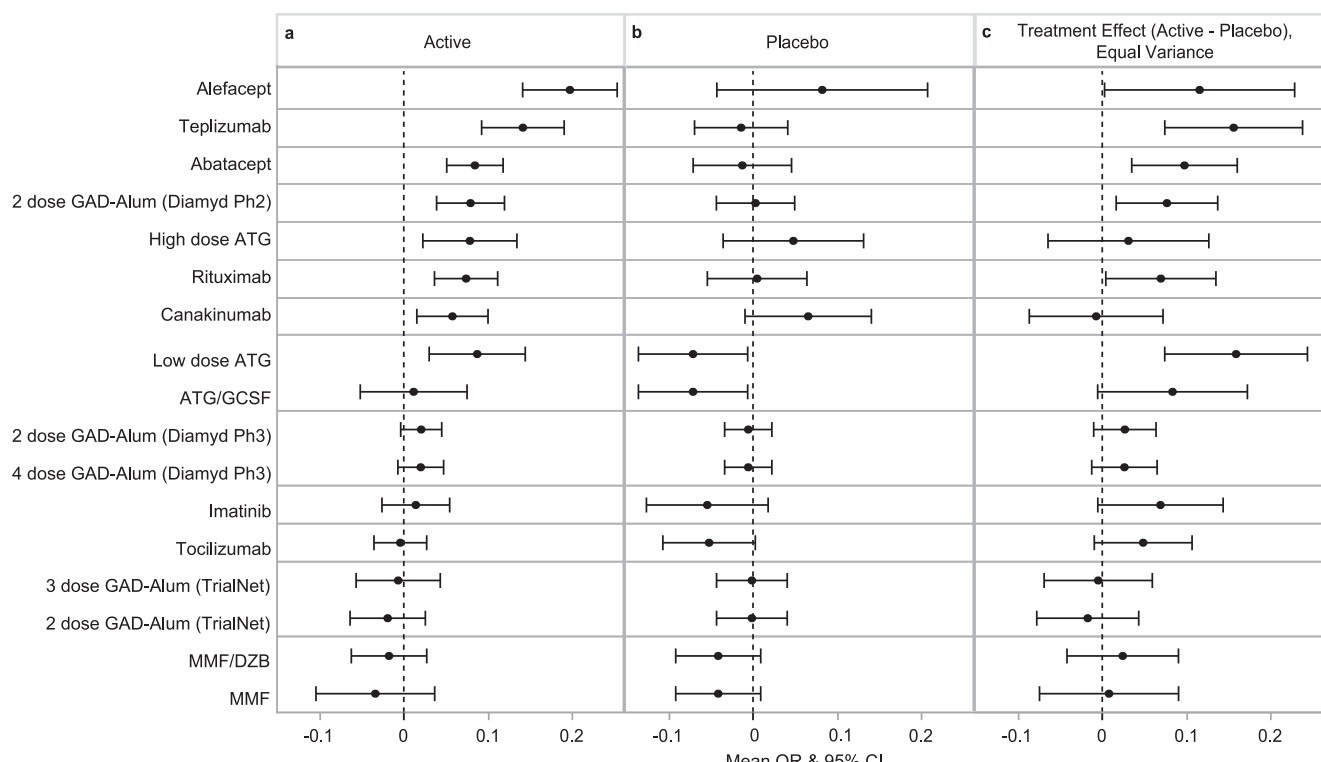

**Fig. 5 | Use of QR to interpret across clinical trials.** Mean QR ± 95% confidence interval (CI) for (**a**) active treatment, (**b**) control/placebo group, and (**c**) the difference between treatment arms for each trial determined from two-tailed *t*-test. *N* for each treatment group is reported in Table 1.

suggest that with these caveats in mind, data garnered from single arm trials could inform decisions about whether a therapeutic approach merits further testing in subsequent gold standard phase 3 placebo-controlled randomized trials[37,38].

Furthermore, using a QR outcome allows for adaptive study designs of multiple active agents, as we found that a mean QR value above zero in the treatment arm at six months after randomization completely predicted the success of all the tested RCTs with a positive outcome at 1 year. A trial with multiple active agents could drop ineffective therapeutic arms at six months. Using QR could enable shorter clinical trials, which would reduce burden on study staff and participants, reduce cost, and reduce the time that participants in the active arm are exposed to ineffective therapies. By pre-specifying a QR threshold of interest at an early timepoint, adaptive re-randomization designs, such as the sequential parallel comparison design (SPCD; Supplemental Fig. 4)[39], are also feasible. This design identifies placebo participants with a QR value below zero early in a trial, and re-randomizes those individuals to treatment or placebo, allowing a larger number of participants to potentially benefit from therapy. QR can also enable enrollment of individuals outside the traditionally used period of 3 months post-diagnosis in new-onset T1D trials, as it can reliably predict a 2-year outcome when a baseline timepoint is >6 months from diagnosis.

Assessing treatment effect as the difference in QR between treatment and control arms may alter interpretation of prior trial results. In addition, it can aid in prioritizing therapies for further study. Since there were similarities between the 13 trials with respect to baseline C-peptide and age, and many of the trials used baseline-adjusted ANCOVA models, it is not surprising that using QR to express the trial result is similar to that seen in published reports; that is, the teplizumab, abatacept, rituximab, and low dose ATG trials all demonstrate that the QR of the actively treated group is higher than that of the control group. However, while the published primary outcome of the ITN alefacept trial was negative, when

considering the outcomes of the active treatment arms for each trial, teplizumab and alefacept both stand out as therapies with the greatest QR values, which suggests both therapies (or similar drugs) are worth pursuing. Furthermore, while the published results of two trials using ATG differ (ITN higher dose trial being negative and TrialNet lower dose trial being positive), the QR point estimates of the active arms in each of these trials are similar, indicating that the differences in clinical trial outcomes reported were perhaps impacted by differences in the placebo arms rather than differences in efficacy between the two doses.

The canakinumab trial exemplifies the risks of comparison to a small control cohort. The originally reported canakinumab negative trial result had a detrimental impact on future studies; despite pre-clinical and mechanistic data suggesting a role of IL-1 in T1D[40–43], there has been reluctance to test this type of therapy further. However, we show here that the QR of individuals in the canakinumab-treated arm was positive, suggesting therapeutic effect. Our interpretation of the data indicates that the negative result in the originally published trial was due to the small number of placebo participants who performed much better than expected. Of note, in a retrospective analysis of the original study, Bundy et al. addressed this issue by comparing canakinumab-treated individuals to a larger placebo group, and also concluded that canakinumab may be effective[36]. While not negating the results of the original RCT, the integration of large amounts of historical data here provides added context to robustly interpret studies.

Perhaps the most powerful use of QR is its ability to determine the extent to which an individual responded to therapy. Participants are typically informed of clinical trial results with information about their own insulin secretion and the mean values for treatment and control groups. However, explaining what this means can be challenging as the relationship between a given C-peptide value or change in C-peptide with long term clinical outcomes is not known. QR, in contrast, allows for standardized, subject-specific estimates to be provided to each

participant; study staff can describe the probability that the participant did better or worse than expected while on treatment (i.e., a responder or non-responder).

QR is also an improvement over previously used responder/non-responder definitions. Incorporating historical data via QR provides a greater level of certainty when identifying treatment responders. Standardized estimates based on historical placebo data can be used to understand the probability of observing a specific QR value in the absence of a treatment effect. Higher QR values are associated with increased confidence that an individual's response is related to treat-

ment. In the absence of a QR framework, we would be less certain about these predictions at both the individual and group level. In trials with two or more active treatments with differing mechanisms of action, the QR can be used as a standardized instrument to discriminate biomarkers hypothesized to be causally related to treatment with the objective of personalizing immune therapies to specific endotypes[44].

QR is particularly useful for a more principled analysis of mechanistic data seeking to explain whether a mechanistic marker lies in the hypothesized causal pathway for the therapy. This concept is exemplified by analysis of exhausted T cells in individuals treated with teplizumab. The increase in these cells post-therapy is associated with C-peptide, and there is also a relationship between T cell exhaustion and age, consistent with previous reports[45–47]. This suggests that understanding the causal pathway between teplizumab therapy and the induction of exhausted T cells must consider age as a factor[48], while also helping the field to consider the general phenomenon of why children may be more likely to respond to therapy.

Using data from 13 clinical trials and more than 1,300 participants, we demonstrate the significant value of using QR to advance the field of DMT in T1D. Our study serves as an example for applying the QR approach in other diseases that lack clarity in defining responders to therapy, for comparing the effectiveness of different therapies, and for understanding causal pathways in disease. Our analysis shows that the QR metric of insulin secretion measured by C-peptide is clinically and scientifically meaningful, objective, predictable, and standardized across individuals and cohorts, thus accelerating and aiding in interpretation of trials and providing a framework for precision medicine in T1D.

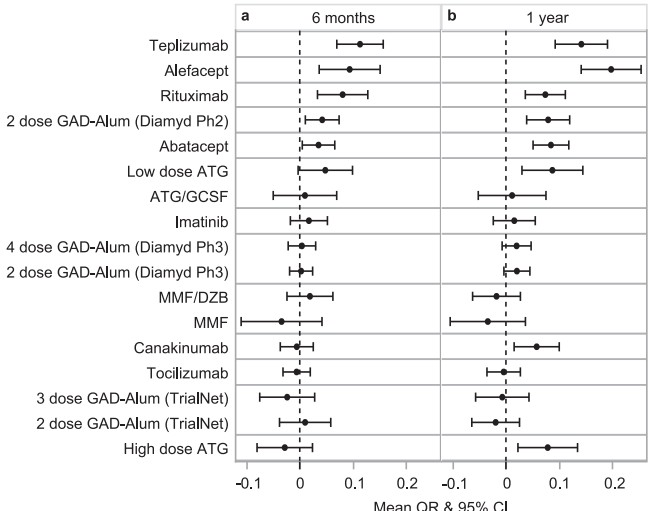

**Fig. 6 | Effect of therapy determined at 6 months after randomization.** Mean QR ± 95% confidence interval (CI) at (**a**) 6 months and (**b**) 12 months. All treatment arms with mean QR ± 95% CI above 0 at 6 months were also above 0 at 1 year. Since the QR metric is based on a linear model, where baseline is predictive of 1 year C-peptide, QR at interim timepoints was computed by deriving the expected C-peptide values at specified timepoints from the original QR equation, and determining the difference between the expected and observed values. At 6 months: teplizumab $n = 51$, alefacept $n = 31$, rituximab $n = 52$, 2 dose GAD-Alum (Diamyd Ph2) $n = 35$, abatacept $n = 67$, low-dose ATG $n = 29$, ATG/GCSF $n = 28$, imatinib $n = 43$, 4 dose GAD-Alum (Diamyd Ph3) $n = 109$, 2 dose GAD-Alum (Diamyd Ph3) $n = 107$, MMF/DZB $n = 39$, MMF $n = 28$, canakinumab $n = 47$, tocilizumab $n = 87$, 3 dose GAD-Alum (TrialNet) $n = 44$, 2 dose GAD-Alum (TrialNet) $n = 46$, high-dose ATG $n = 36$. Reference Table 1 for n at 12 months.

## Methods

### Datasets

De-identified data were obtained from 13 clinical trials of DMT in individuals with recently diagnosed T1D (Table 1). These include six studies conducted by Diabetes TrialNet (TrialNet.org), an NIH-sponsored clinical trial network: MMF/DZB (TN02 NCT00100178[12]), Rituximab (TN05 NCT00279305[11]), GAD-Alum (TN08 NCT00529399[7]), Abatacept (TN09 NCT00505375[6]), Canakinumab (TN14 NCT00947427[8]), Low Dose ATG/ GCSF (TN19 NCT02215200[9]); four studies conducted by the NIH funded Immune Tolerance Network (ITN; Immunetolerance.org): Teplizumab (AbATE NCT02067923[13]), Alefacept (T1DAL NCT00965458[14]), ATG (START NCT00515099[18]), Tocilizumab (EXTEND NCT02293837[15]); one investigator-initiated study sponsored by JDRF, Imatinib/Gleevec

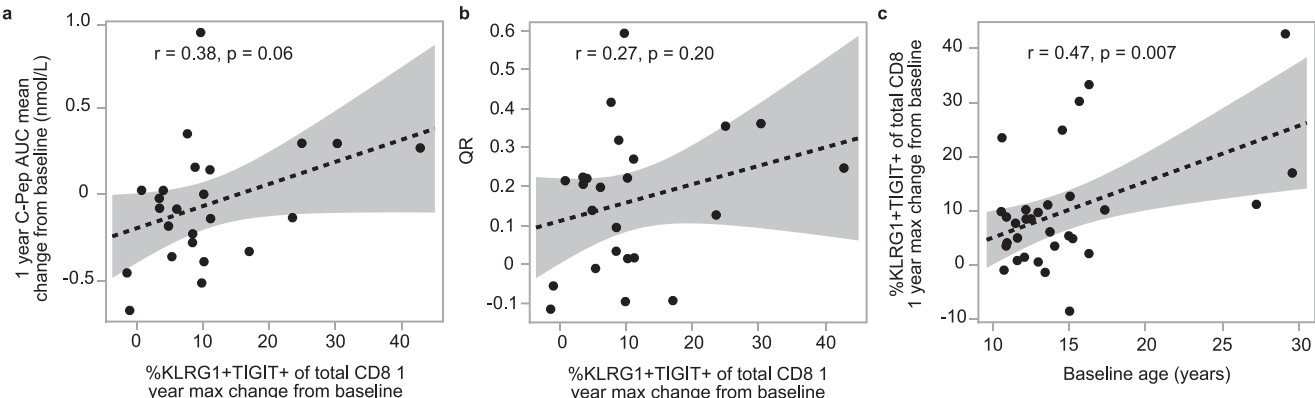

**Fig. 7 | %KLRG1 + TIGIT+ of CD8 T cells associated with age and outcome of teplizumab therapy.** C-peptide and %KLRG1 + TIGIT+ of CD8 T cells (exhausted T cells) in actively treated individuals ($n = 31$) from ITN trial of teplizumab. Dashed line and shaded region indicate linear regression line and 95% confidence intervals. **a** Relationship between increase of exhausted T cells and one-year C-peptide;

Pearson correlation ($r$) of 0.38, $p = 0.06$. **b** Relationship between exhausted T cells and QR; $r = 0.27$, $p = 0.20$. **c** Correlation between exhausted T cells and age; $r = 0.47$, $p = 0.007$. T cell exhaustion was quantified as the maximum change from baseline at any point during the one year time period after treatment initiation.

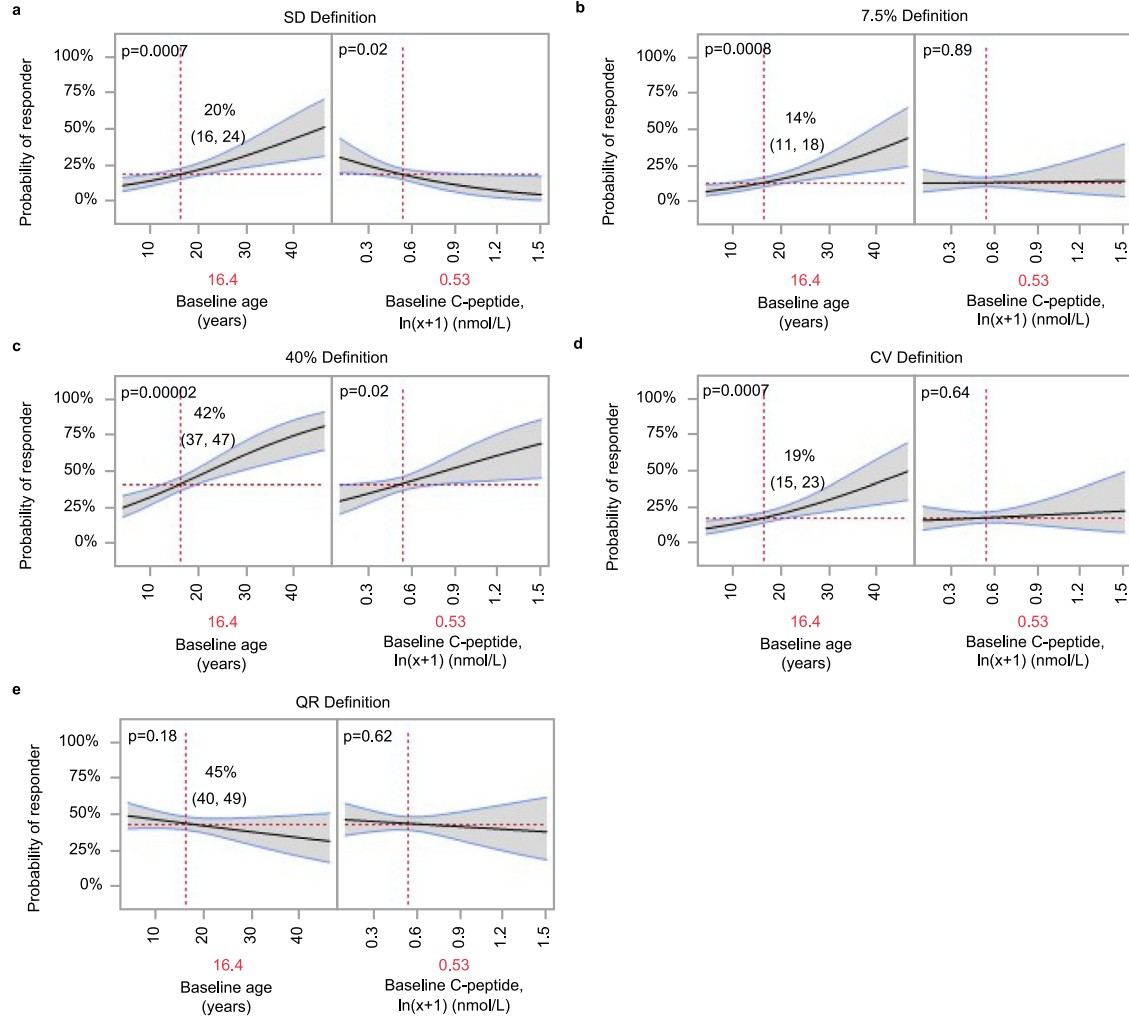

**Fig. 8 | Proportion of placebo/control individuals meeting responder definitions across trials.** Logit-link binomial generalized linear models fit in placebo/control ($n = 448$) participants, where baseline age (years) and baseline C-peptide ($\ln(x + 1)$, nmol/L) are used to predict responder outcomes for QR-based responder definition and responder definitions reported in the literature. Mutually adjusted estimates for each predictor on the x-axis correspond to model predictions (mean shown by black regression line) and 95% confidence intervals on the y-axis (gray shading with blue bounds). The probability of a responder is benchmarked (indicated by vertical dashed red lines and annotated in red on x-axis) to an individual of average age (16.4 years) and baseline C-peptide ($\ln(Cp_0 + 1)$ of 0.53 or 0.70 nmol/L); QR of 0.039. The probability (horizontal dashed red line) and 95% confidence

interval of a placebo participant with these baseline characteristics being identified as a responder are annotated where dashed red lines intersect. *P*-values annotated for likelihood ratio tests of fixed effects. No multiple comparisons adjustment was done. **a** Inter-test SD definition: responders identified as those whose C-peptide change from baseline is nonnegative or negative but no more than 1 inter-test SD of 0.087 nmol/L below baseline (from refs. 25, 35). **b** 7.5% definition: responders defined as those with C-peptide decline of no more than 7.5% below baseline[34]; **c** 40% definition: responders are those with <40% loss of baseline C-peptide[13]; **d** CV definition: responders are those with nonnegative change from baseline or negative but coefficient of variation (CV) < 0.097 (median CV from refs. 25, 35). **e** QR > 0 definition: responder is defined as positive QR.

(NCT01781975[10]); and two industry led studies, Diamyd Therapeutics AB: Phase 2 (NCT00435981[17]) and Phase 3 (NCT00723411[16]) GAD-Alum.

## Statistical methods

The ANCOVA model developed by Bundy and Krischer[27], using data from recently diagnosed T1D individuals, $QR_i = \ln(Cp_{1year,i} + 1) - 0.812 \cdot \ln(Cp_{0,i} + 1) - 0.00638 \cdot Age_i + 0.191$, was used to compute the individual's QR, where $Cp_{0,i}$ and $Cp_{1year,i}$ represent 2-hour C-peptide AUC mean (AUC divided by 120 min, in nanomoles per liter) at baseline and one year post treatment, respectively; $Age_i$ is the age at study entry, in years. Since the model assumes a linear relationship between baseline and 1 year C-peptide, we additionally computed QR at 3, 6, and 9 months post-randomization by deriving the expected C-peptide values at these timepoints from the original QR equation, and determining the difference between the expected and observed values at each timepoint. Other variables were considered for inclusion in the original model and were found not to improve model fit[27,36].

To validate the QR method, we tested the model performance by applying the published ANCOVA model to data from eight new studies not used for the development of QR (Fig. 1). The Kolmogorov-Smirnov two-sample test was used to compare QR distributions between the development ($n = 5$ studies) and validation ($n = 8$ studies) cohorts. In addition, an ANCOVA model was developed using all control participants to assess the association between actual 1 year C-peptide AUC mean values and predicted values from both the formula derived from the ANCOVA developed from our dataset and the published QR formula.

Participants were classified as either active treatment or placebo/control. Two-sample, two-sided *t*-tests were used for comparison of means between groups. For responder analyses, participants were dichotomized based on historical thresholds from the literature used to define responders and non-responders, and using a QR responder definition, where responders are individuals with positive QR and non-responders are individuals with negative

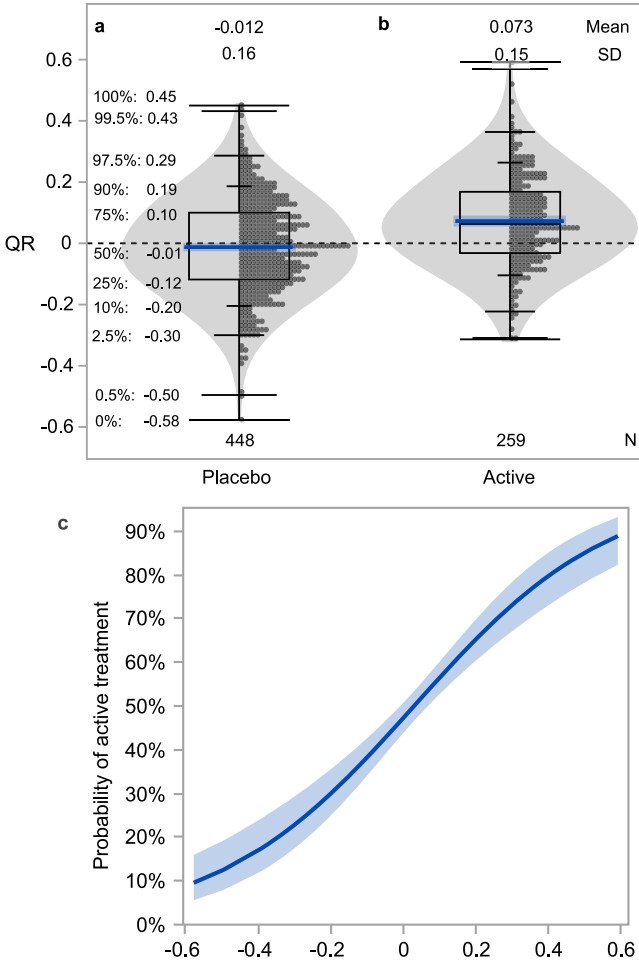

**Fig. 9 | QR values for control/placebo and actively treated individuals in trials reported with positive outcomes inform ability to distinguish actively treated from placebo. a** QR values for control/placebo (*n* = 448) participants with corresponding percentiles. Blue line and shaded region indicate mean and standard deviation (SD). **b** QR values for active treatment individuals in trials reported with positive outcome (*n* = 259). Blue line and shaded region indicate mean and SD. **c** Probability (bold blue line) of identifying actively treated individuals with 95% confidence intervals (blue shaded region). The distribution shown among participants treated with drugs from positive trials (**b**) or placebos (**a**) was used in a logistic regression model with active treatment or placebo as the outcome. To account for imbalance between active (*n* = 259) and placebo (*n* = 448) groups, observations were weighted so that active and placebo participants were equally represented in producing the probability curve (**c**) shown. Active treatment observations were weighted more heavily than placebo observations; weights were determined as the target proportion divided by the actual proportion, where the target proportion is equivalent to the proportion of the dataset composed of placebo participants (i.e., the larger group, 63%). Quantile box plots (**a, b**) show the distribution of data; whiskers from top to bottom indicate percentiles: 0, 0.5, 2.5, 10, 90, 97.5, 99.5, 100. The box is bounded at the 25th and 75th percentiles (interquartile range) and horizontal line within box indicates median 50th percentile. Percentiles are also annotated. Violin plots (**a, b**) visualize data distribution.

QR. Generalized linear models with a binomial distribution and a logit link were fit among placebo/control participants, to each responder definition with adjustments for baseline age and baseline C-peptide. For biomarker analyses, Pearson correlations were computed to examine the association of KLRG1 + TIGIT + CD8 + T cells with baseline metrics (age and C-peptide), and with outcome metrics (QR and C-peptide).

Using control group data, additional ANCOVA models were developed to expand the utility of the QR method to different time intervals. Specifically, post-baseline predictions ranging from 3 months to 2 years were created using different baseline reference points and prediction horizons. Analysis was performed using SAS software version 9.4 (SAS Institute Inc., Cary, NC, USA) and JMP Pro 16 (SAS Institute Inc., Cary, NC, USA).

### SAS code utilization
SAS and JSL code are provided at https://github.com/BenaroyaResearch/qr_t1d_metric/. SAS code computes QR and model-predicted C-peptide values at various timepoints using the QR model[27], fits ANCOVA models following the same methodology as the published QR model, and runs t-tests to determine treatment effect using QR. JSL code fits generalized linear models to examine the association between QR and treatment group, and association of baseline metrics (C-peptide and age) with responder status, where responder status determines QR ≥ 0 as responder and QR < 0 as non-responder. A test dataset (SAS dataset) is also provided, which is a one record per subject dataset, including C-peptide AUC mean from 2-h MMTT at all available timepoints (at minimum, baseline and 1 year required), age at screening in years, and treatment group and study. An example output dataset of computed QR and predicted C-peptide values is also provided (SAS and JMP datasets). These codes have been validated and run by multiple analysts and applied to other datasets; they have not been submitted to community commenting.

### Reporting summary
Further information on research design is available in the Nature Portfolio Reporting Summary linked to this article.

## Data availability
All clinical trial source data supporting the findings described in this manuscript are available under controlled access due to data privacy laws. TrialNet clinical trials data are publicly available and can be obtained by application to the NIDDK Central Repository at https://repository.niddk.nih.gov/home/. Immune Tolerance Network clinical trials data are also publicly available at https://www.itntrialshare.org/. Data from the Imatinib study are available from Dr. Stephen Gitelman (stephen.gitelman@ucsf.edu) per data sharing statements from the original publication[10]. Data from the Diamyd Medical GAD-alum phase 2 and phase 3 trials are available upon reasonable request via a data transfer agreement. Requests should be addressed to Anton Lindqvist at anton.lindqvist@diamyd.com.

## Code availability
SAS code supporting all analyses are available at GitHub https://github.com/BenaroyaResearch/qr_t1d_metric/. SAS code is also available from the corresponding author on reasonable request.

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

## Acknowledgements

The authors gratefully acknowledge the access to data from Diamyd trials of GAD-alum from Diamyd Medical and for the Imatinib trial from Principal Investigator Stephen Gitelman, UCSF. Publicly available data was obtained from trials conducted by the Immune Tolerance Network (an international clinical research consortium headquartered at the Benaroya Research Institute and supported by the National Institute of Allergy and Infectious Diseases and JDRF) and from The Type 1 Diabetes TrialNet Study Group. We also thank Anne Hocking and Taylor Lawson (BRI) for thoughtful comments on this manuscript. This analysis was funded in part by JDRF grant 3-SRA-2019-791-S-B (to C.S.) and NIDDK grant 5R03DK127475-02 (to C.S.). TrialNet is a clinical trials network funded by the National Institutes of Health (NIH) through the National Institute of Diabetes and Digestive and Kidney Diseases, the National Institute of Allergy and Infectious Diseases, and The Eunice Kennedy Shriver National Institute of Child Health and Human Development, through the cooperative agreements U01 DK061010, U01 DK061034, U01 DK061042, U01 DK061058, U01 DK085461, U01 DK085465, U01 DK085466, U01 DK085476, U01 DK085499, U01 DK085509, U01 DK103180, U01 DK103153, U01 DK103266, U01 DK103282, U01 DK106984, U01 DK106994, U01 DK107013, U01 DK107014, UC4 DK106993, UC4DK117009. The funders had no role in the conceptualization, design, data collection, analysis, decision to publish, or preparation of the manuscript.

## Author contributions

Conceptualization: H.T.B., C.J.G., and C.S. Data curation: A.Y. and H.T.B. Formal analysis: A.Y., H.T.B., and C.O. Visualization: A.Y. and H.T.B. Funding acquisition: C.J.G. and C.S. Writing—original draft: C.J.G., A.Y., H.T.B., and C.S. Writing—review & editing: C.J.G., A.Y., H.T.B., C.O., S.L., and C.S.

## Competing interests

The authors declare no competing interests.
