## [Peer Review File · Nature Communications]

A Standardized Metric to Enhance Clinical Trial Design and Outcome Interpretation in Type 1 DiabetesREVIEWER COMMENTS

Reviewer #1 (Remarks to the Author):

The quantitative response (QR) metric proposed in this paper is specific for diabetes trials and cannot be directly used in trials of other diseases. Therefore 'diabetes' needs to be added to the title somewhere. The current title overstates the scopes of this paper.

In Figure 2, the authors showed standard deviation (SD) of AUC and QR and said QR has reduced SD than AUC. However, SD is scale dependent and thus not comparable between different outcomes. For example, the outcome length measured in meter and feet will have different SD and a smaller SD does not imply a better measure.

There are a total of 10 figures/tables. This needs to be reduced.

The advantages of a simple outcome such as change of Cp at 1 year from baseline is that its meaning is straightforward. With QR, what does the value mean? E.g., a QR value of 0.1 means what?

QR is calculated from a formula. How valid is this formula? Should other factors (besides age) be included in this formula?

Reviewer #2 (Remarks to the Author):

Description

This paper evaluates the use of a quantitative response (QR) metric as an endpoint in trials of type I diabetes (T1D) and advocates for the use of similar type metrics in clinical trials as a means to improve efficiency, interpretability, and statistical power. The metric was obtained from an analysis of data from five T1D trials, and is an equation for predicting 1-year C-peptide AUC levels using baseline C-peptide and age as predictor variables. The QR metric is the observed 1-year C-peptide value minus the predicted value for each subject. Use of the metric is illustrated using data from treated and placebo patients participating in 13 T1D

trials, including the five from which the predicting equation was derived. The reduction in variance compared to that from unadjusted outcomes analyzed by the original trial investigators, comparisons with dichotomized measures categorizing patients as responders vs. non-responders, and its use in assessing an immune biomarker to obtain mechanistic insights is also described.

Strong Points

The paper does a very good job demonstrating the advantages of using covariates in clinical trials to improve efficiency relative to unadjusted comparisons when there are strong predictors, particularly in smaller trials. While this is not a new concept, as pointed out by the authors most trialists rely on unadjusted comparisons in RCTs and therefore the points being made here, and they are made very well, are worthy of publication. The argument is enhanced through the use of real data from actual trials. The paper is also clearly written with excellent graphs demonstrating the benefits arising from the QR outcome metric. For example, the standard deviation (SD) of the QR metric is shown to be considerably smaller than the SD of the C-peptide outcome itself | Figure 2B (about a 50% reduction across the 13 trials). The alefacept study is re-analyzed, finding a statistically significant difference using the QR metric while the original analysis was not significant. Figure 6 exhibits the potential benefits of using QR at an earlier time point (6 months rather than 1 year) for interim monitoring in adaptive trials. Another strength of the paper is the immune biomarker analysis, in which it is shown that when assessing the causal pathway between teplizumab therapy and T cell exhaustion, age needs to be taken into account because of the relationship between T cell exhaustion and age.

Major Concerns

While the manuscript has the strengths noted above, a major concern lies in the Discussion. The authors go beyond the efficiency argument and make the rather strong statements that “Utilizing QR as an outcome measure implies comparison of a treatment arm to this large

number of historical controls, and may obviate the need for a contemporaneous control group”; “The QR framework achieves these advantages by leveraging large amounts of historical data to create “synthetic controls” to test promising interventions in phase 2 trials”; and “Because QR has been defined for hundreds of untreated individuals, this increased confidence in defining treatment response can also enable design of clinical trials that use only active treatments.” This is a very problematic assertion that such metrics can eliminate the need for randomized comparisons, and is actually contradicted by the cited data. The authors note that, in the cnaKnumab trial (which was negative), the distribution of QR of the active arm (figure 5A) was shifted to the right, with a mean significantly greater than 0 (Figure 5A). They argue that this “suggests a positive effect of this therapy on C-peptide.” However the QR distribution in the placebo arm is also shifted to the right (Figure 5B). (The fact that the 95% CI just crosses 0 should not be used to negate my point.) The authors draw the conclusion, further elaborated upon in the Discussion, that “the lack of statistical significance between the groups may be driven by higher-than-expected C-peptide response in the 22 individuals in the placebo arm of the study.” But an alternative explanation is that this trial selected patients for enrollment that were at lower risk than the populations from which the QR prediction equation was derived. This is precisely the reason for having contemporary, randomized controls, and it is unfortunate that the authors have gone beyond the efficiency argument and leapt to the conclusion that randomized controls might be dispensed with if one makes use of prognostic covariates.

A second concern is that the paper includes the five trials from which the QR predictor was developed (Bundy et al, 2020). The paper would be strengthened, I think, if it simply cited that work and then utilized only the validation cohort of eight trials. Based on Figure 2, the efficiency gains would still be apparent, and the presentation would not be clouded with the concern that the value of the QR metric was partly assessed using data on which the metric itself was derived, which is known to produce an optimistic bias.

I also cannot help but wonder how the QR metric would compare with performing an ANCOVA within each trial. It is true that the QR equation was obtained from a large number of patients, but it would be interesting to see if, in fact, the two approaches lead to very similar findings. Some relevant literature on surrogate outcomes, as well as the FDA

guidance on “Adjusting for Covariates in Randomized Clinical Trials for Drugs and Biologics with Continuous Outcomes” should be cited and discussed a bit. Finally, to make a more general case for standardized metrics, potential covariates/predictive models in other diseases should be offered.

Minor Issues

Line 40: There is no “average treated individual” or “average control individual”. Suggest rewording.

Lines 43-44: A downside of covariate adjustment is the potential to magnify the treatment effect by fitting multiple models and selecting the one with the smallest p-value (see Beach and Meier, *Controlled Clinical Trials*, 1989). Thus specification in advance of the covariates to be incorporated is generally recommended and mention of this issue is warranted. One advantage here is that baseline C-peptide and age have been established as strong predictive covariates in T1D studies using C-peptide as the outcome. But this will not apply to other diseases and situations.

Line 90: Figure 1 shows sample sizes of 162 and 286. Where do the >1,000 degrees of freedom come from?

Line 135: Pointing out the predictive value of the 6, 12, and 18 month data is important, but in terms of predicting longer-term outcomes beyond 1 year, only the 6-month R² is relevant.

Lines 158-169: The dosing example is confusing. What is the conclusion here? Is it that the higher dose may still be more effective than the lower dose? If so, this relies on cross-study comparisons, which is problematic.

Line 182: Presenting the correlation between age and C-peptide would be helpful.

Line 372: One can't help but wonder why these diabetes trials were all so small. It's a common disease, so why weren't the sample sizes larger?

Line 644: For the development cohort, the distribution must be centered at 0, right, since the mean is based the sum of residuals?

Figure 7A and 7B: The authors might want to reverse x and y axis to match the y-axis of Figure C.

Line 702: Is this generalized linear models or just logistic regression?

Line 727: Please clarify the weighting.

Figure 9: Would an ROC-curve type analysis be beneficial here?

Reviewer #3 (Remarks to the Author):

The authors re-examine the results of Phase 2 trials aimed at preserving endogenous insulin secretion in individuals at high risk of progressing to type 1 diabetes suggesting a recently-described potential refinement of the accepted (2 hour C-peptide AUC) regulatory outcome measure (named "QR") which adjusts for baseline values and age making use of previous data on C-peptide trajectories - hence reporting the difference between the observed and predicted values. The authors demonstrate that in some trials random "positive" effects in the control group can be seen to mask potentially useful treatment effects when reviewed in the context of other control groups from similar trials.

This approach may also offer a way of reducing sample size for Phase 2 proof of concept trials, better predicting "responders," and moving to platform type trials.

Comments

- 1) The arguments set out are overall convincing
- 2) Can the authors comment on the reason for quite large mean increases in C-peptide secretion in control groups of the studies with teplizumab and canakinumab?

3) Is there a possibility that this approach could be over-sensitive i.e. allow weaker candidate molecules to progress to Phase 3? It should be emphasised that the ultimate outcome measure is delaying the onset of clinical type 1 diabetes by a meaningful length of time.

4) The comments regarding using similar approaches in other conditions seem a little off topic and could be toned down - adjusting for baseline value is quite widely performed for other outcome measures.

5) The authors should clarify that their comments on a potential lack of requirement for contemporaneous controls are restricted to Phase 2 trials

6) The individual who originally described the QR measure is a member of the same consortium - is there a reason why he or she is not included here in the author group?

We thank all of the reviewers for their thoughtful comments, and we provide a point-by-point response below. Please note that all line numbers refer to the marked-up/tracked-changes version of the revised manuscript which includes sentences with “strike-through” notation to highlight sections that have been removed from the revised version.

REVIEWER COMMENTS

Reviewer #1 (Remarks to the Author):

a. The quantitative response (QR) metric proposed in this paper is specific for diabetes trials and cannot be directly used in trials of other diseases. Therefore ‘diabetes’ needs to be added to the title somewhere. The current title overstates the scopes of this paper.

RESPONSE: Thank you for the comment. We have added type 1 diabetes to the title as suggested (lines 2-3).

b. In Figure 2, the authors showed standard deviation (SD) of AUC and QR and said QR has reduced SD than AUC. However, SD is scale dependent and thus not comparable between different outcomes. For example, the outcome length measured in meter and feet will have different SD and a smaller SD does not imply a better measure.

RESPONSE: We appreciate the reviewer’s point that generically a reduced SD is not a “better” measure; however, the reduction in SD using the QR metric impacts clinical trial design as it increases power and therefore the potential for reducing sample size.

c. There are a total of 10 figures/tables. This needs to be reduced.

RESPONSE: We appreciate this perspective. However, the editor informed us that we are within the limit for figures/tables and we respectfully prefer to keep the existing figures and tables.

d. The advantages of a simple outcome such as change of Cp at 1 year from baseline is that its meaning is straightforward. With QR, what does the value mean? E.g., a QR value of 0.1 means what?

RESPONSE: We appreciate this comment. Of course, any metric can be challenging to interpret clinically. The QR value of 0.1 means that an individual had insulin secretion above what would have been expected for a person of their age and baseline insulin secretion. Interestingly, it is the change in C-peptide itself that does not have a straightforward meaning. There is currently no gold standard C-peptide value associated with clinical outcomes, or that can indicate success of disease modifying therapy (lines 62-66). Though individual C-peptide data after a trial is routinely shared with study participants, investigators are hard pressed to explain what this means. In contrast, a QR value can provide context about how much better or worse an individual did than they were expected to do. We agree this is an important point to make and have edited the discussion to address this concept (see discussion lines 357-359).

e. QR is calculated from a formula. How valid is this formula? Should other factors (besides age) be included in this formula?

RESPONSE: We appreciate the reviewer’s question as it highlights a clarification we should make. Since QR accounts for ~half of the variance in the outcome, the reviewer is right to point out that other factors beyond age and baseline insulin secretion must play a role in the loss of insulin secretion downstream. The original papers describing this model (Bundy et al, Endocrinol Diabetes Metab 2020 and Bundy et al Diabetes Metab Res Rev 2016) evaluated other covariates for statistical significance - such as HbA1c, sex, and BMI. The authors noted that though HbA1c was statistically significant, it did not improve model fit due to collinearity with baseline C-peptide. For this

paper, we demonstrate an application of QR and focus on validation of the previously published formula, rather than deriving a new and/or improved equation. It is possible that future work will identify other important variables; in which case the model can be updated with new information. We have added a statement to the methods that the analysis of other potential variables was included in the original model development (lines 414-415).

Reviewer #2 (Remarks to the Author):

Description

This paper evaluates the use of a quantitative response (QR) metric as an endpoint in trials of type I diabetes (T1D) and advocates for the use of similar type metrics in clinical trials as a means to improve efficiency, interpretability, and statistical power. The metric was obtained from an analysis of data from five T1D trials, and is an equation for predicting 1-year C-peptide AUC levels using baseline C-peptide and age as predictor variables. The QR metric is the observed 1-year C-peptide value minus the predicted value for each subject. Use of the metric is illustrated using data from treated and placebo patients participating in 13 T1D trials, including the five from which the predicting equation was derived. The reduction in variance compared to that from unadjusted outcomes analyzed by the original trial investigators, comparisons with dichotomized measures categorizing patients as responders vs. non-responders, and its use in assessing an immune biomarker to obtain mechanistic insights is also described.

Strong Points

The paper does a very good job demonstrating the advantages of using covariates in clinical trials to improve efficiency relative to unadjusted comparisons when there are strong predictors, particularly in smaller trials. While this is not a new concept, as pointed out by the authors most trialists rely on unadjusted comparisons in RCTs and therefore the points being made here, and they are made very well, are worthy of publication. The argument is enhanced through the use of real data from actual trials. The paper is also clearly written with excellent graphs demonstrating the benefits arising from the QR outcome metric. For example, the standard deviation (SD) of the QR metric is shown to be considerably smaller than the SD of the C-peptide outcome itself (Figure 2B (about a 50% reduction across the 13 trials)). The alefacept study is re-analyzed, finding a statistically significant difference using the QR metric while the original analysis was not significant. Figure 6 exhibits the potential benefits of using QR at an earlier time point (6 months rather than 1 year) for interim monitoring in adaptive trials. Another strength of the paper is the immune biomarker analysis, in which it is shown that when assessing the causal pathway between teplizumab therapy and T cell exhaustion, age needs to be taken into account because of the relationship between T cell exhaustion and age.

Major Concerns

a. While the manuscript has the strengths noted above, a major concern lies in the Discussion. The authors go beyond the efficiency argument and make the rather strong statements that “Utilizing QR as an outcome measure implies comparison of a treatment arm to this large number of historical controls, and may obviate the need for a contemporaneous control group”; “The QR framework achieves these advantages by leveraging large amounts of historical data to create “synthetic controls” to test promising interventions in phase 2 trials”; and “Because QR has been defined for hundreds of untreated individuals, this increased confidence in defining treatment response can also enable design of clinical trials that use only active treatments.” This is a very problematic assertion that such metrics can eliminate the need for randomized comparisons, and is actually contradicted by the cited data. The authors note that, in the cpanumab trial (which was negative), the distribution of QR of the active arm (figure 5A) was shifted to the right, with a mean significantly greater than 0 (Figure 5A). They argue that this “suggests a positive effect of this therapy on C-peptide.” However the QR distribution in the placebo arm is also shifted to the right (Figure 5B). (The fact that the 95% CI just crosses 0 should not be used to negate my point.) The authors draw the conclusion, further elaborated upon in the Discussion, that “the lack of statistical significance between the

groups may be driven by higher-than-expected C-peptide response in the 22 individuals in the placebo arm of the study.” But an alternative explanation is that this trial selected patients for enrollment that were at lower risk than the populations from which the QR prediction equation was derived. This is precisely the reason for having contemporary, randomized controls, and it is unfortunate that the authors have gone beyond the efficiency argument and leapt to the conclusion that randomized controls might be dispensed with if one makes use of prognostic covariates.

RESPONSE: We agree with the reviewer that the use of historical or synthetic controls has its own risks and acknowledge that examples of this were embedded in our manuscript. We fully agree with the reviewer that the RCT remains the gold standard; but these have their own limitations, particularly when the sample sizes are small. For example, we agree with the reviewer regarding the canakinumab trial that the RCT results are appropriately taken as face value. Here, we seek to provide additional context to interpret the published trial results.

There may be pragmatic or other reasons that an RCT is not always feasible. The conceptual purpose of an early phase clinical trial is to determine a yes/no decision about conducting larger, more definitive studies. We only wish to indicate that if one is not willing to or able to conduct an RCT in such early studies, the use of QR can allow for single arm trials – if and only if one acknowledges the limitations to such an approach. We have significantly revised the discussion to reflect the reviewer’s concerns (lines 304-307, 309-315, and 369-371).

b. A second concern is that the paper includes the five trials from which the QR predictor was developed (Bundy et al, 2020). The paper would be strengthened, I think, if it simply cited that work and then utilized only the validation cohort of eight trials. Based on Figure 2, the efficiency gains would still be apparent, and the presentation would not be clouded with the concern that the value of the QR metric was partly assessed using data on which the metric itself was derived, which is known to produce an optimistic bias.

RESPONSE: We thank the reviewer for this comment and appreciate this concern. To address the issue raised, we fit an ANCOVA model to the placebo participants in only the validation cohort (8 trials, n=286). Using the predicted 1-year C-peptide values from this model, we then computed a “new” QR, which is shown on the y-axis of the figure presented at right. The published QR (x-axis) and this “new” QR are highly correlated (r=0.99) in both the validation (n=8 studies) and development (n=5 studies) cohorts. Based on this new finding

– and the goal of our work which was to comprehensively apply QR to all available trials - we continue to use data from all 13 trials for many analyses. We have also added lines 93-99 to the text describing these findings and added the figures above as new panels of Supplemental Figure 1 (referenced in lines 98-99, 107-108).

c. I also cannot help but wonder how the QR metric would compare with performing an ANCOVA within each trial. It is true that the QR equation was obtained from a large number of patients, but it would be interesting to see if, in fact, the two approaches lead to very similar findings. Some relevant literature on surrogate outcomes, as well as the FDA guidance on “Adjusting for Covariates in Randomized Clinical Trials for Drugs and Biologics with

Continuous Outcomes” should be cited and discussed a bit. Finally, to make a more general case for standardized metrics, potential covariates/predictive models in other diseases should be offered.

RESPONSE: Many thanks for these comments. First, we now discuss the referenced FDA guidance in the introduction and have added a reference to the existing literature on this topic, as suggested (lines 40-45 and new references 1 and 2).

Secondly, since the QR is based on an ANCOVA model, it is expected that the treatment effect determined from either method will be similar and this is demonstrated in the below table for the reviewer. Of course, QR has the advantage of being a standardized metric and is more easily interpretable.

Lastly, other reviewers suggested that we focus on type 1 diabetes rather than making a more general case for our approach and thus we have deleted this concept. However, we are open to thoughts from the editor and reviewers about this issue, and therefore have mentioned it in our cover letter to the editor.

Treatment Group	ANCOVA 1 year C-pep difference (95%CI), Active-Placebo (nmol/L)	QR difference (95%CI), Active-Placebo
Teplizumab	0.162 (0.072,0.259)	0.156 (0.074,0.238)
2 dose GAD-Alum (Diamyd Ph2)	0.083 (0.016,0.154)	0.077 (0.016,0.137)
2 dose GAD-Alum (Diamyd Ph3)	0.031 (-0.007,0.07)	0.026 (-0.011,0.064)
4 dose GAD-Alum (Diamyd Ph3)	0.029 (-0.009,0.068)	0.026 (-0.013,0.065)
Tocilizumab	0.047 (-0.013,0.109)	0.048 (-0.01,0.106)
Imatinib	0.067 (-0.007,0.148)	0.07 (-0.003,0.144)
High dose ATG	0.034 (-0.061,0.138)	0.031 (-0.065,0.126)
Alefacept	0.114 (-0.015,0.258)	0.115 (0.002,0.229)
DZB & MMF	0.024 (-0.045,0.097)	0.024 (-0.043,0.09)
MMF	0.008 (-0.067,0.089)	0.008 (-0.075,0.09)
Rituximab	0.067 (-0.001,0.14)	0.069 (0.004,0.135)
2 dose GAD-Alum (TrialNet)	-0.022 (-0.081,0.042)	-0.018 (-0.079,0.043)
3 dose GAD-Alum (TrialNet)	-0.002 (-0.064,0.063)	-0.005 (-0.07,0.059)
Abatacept	0.099 (0.032,0.17)	0.097 (0.035,0.16)
Canakinumab	-0.017 (-0.092,0.064)	-0.008 (-0.087,0.072)
ATG & GCSF	0.086 (-0.004,0.185)	0.083 (-0.006,0.172)
Low dose ATG	0.17 (0.074,0.275)	0.159 (0.074,0.244)

Minor Issues

Line 40: There is no “average treated individual” or “average control individual”. Suggest re-wording.

RESPONSE: We have revised throughout as suggested by the reviewer, including lines 230-31, 233-234.

Lines 43-44: A downside of covariate adjustment is the potential to magnify the treatment effect by fitting multiple models and selecting the one with the smallest p-value (see Beach and Meier, Controlled Clinical Trials, 1989). Thus specification in advance of the covariates to be incorporated is generally recommended and mention of this issue is warranted. One advantage here is that baseline C-peptide and age have been established as strong

predictive covariates in T1D studies using C-peptide as the outcome. But this will not apply to other diseases and situations.

RESPONSE: We completely agree with the reviewer and have added this important concept to the introduction (lines 40 and 42-45) and discussion (lines 280-81, 288-289).

Line 90: Figure 1 shows sample sizes of 162 and 286. Where do the >1,000 degrees of freedom come from?

RESPONSE: Thank you for catching this mistake on our part. We have corrected lines 103-104 to now say: “ $p=0.43$, two-sample t-test [$t=0.8$, $DF=346$]” and updated line 618-623 and 712-715 in Figure 1 legend.

Line 135: Pointing out the predictive value of the 6, 12, and 18 month data is important, but in terms of predicting longer-term outcomes beyond 1 year, only the 6-month R² is relevant.

RESPONSE: We thank the reviewer for the opportunity to clarify why we evaluated the utility of QR at different time points. Most “new onset” trials enroll individuals within 3 months of diagnosis. This by definition excludes individuals who could potentially benefit from disease modifying therapy if delivered outside this time window. We thus illustrate that if entry criteria included enrollment at 6, 12, or even 18 months from onset, QR would be an appropriate measure to assess outcome at 2 years. We have revised the text for clarity (lines 147-152).

Lines 158-169: The dosing example is confusing. What is the conclusion here? Is it that the higher dose may still be more effective than the lower dose? If so, this relies on cross-study comparisons, which is problematic.

RESPONSE: We appreciate the reviewer’s comment. We are making no conclusion that the higher dose of ATG is more effective than the lower dose. The differences in the outcomes of the two RCTs using ATG have been ascribed to one study using a different dose than the other. Here we are suggesting that dose may not be the only difference between the trials, highlighting that the outcomes may alternatively be attributed to the differing behavior of the placebo groups in the trials. We have revised the text for clarity (lines 185-187).

Line 182: Presenting the correlation between age and C-peptide would be helpful.

RESPONSE: We appreciate this idea from the reviewer. We analyzed the data, and the 1-year change in C-peptide is not strongly correlated with age; $r=0.06$, $p=0.69$. With such a low correlation, we feel that an additional figure is not helpful. We hope that sharing this information with the reviewer is sufficient to address this concern.

Line 372: One can’t help but wonder why these diabetes trials were all so small. It’s a common disease, so why weren’t the sample sizes larger?

RESPONSE: Most trials of disease modifying therapy in type 1 diabetes are small because enrollment for such studies has always been challenging and the drugs are expensive. Type 1 diabetes, unlike type 2 diabetes, is not a common disease. Type 1 diabetes incidence is 0.3%. Moreover, many incident cases are young children who often are not eligible to enroll in early phase clinical trials.

Line 644: For the development cohort, the distribution must be centered at 0, right, since the mean is based the sum of residuals?

RESPONSE: We thank the reviewer for pointing this out. We agree that the development cohort must center on zero. We have added some clarification (lines 100-103) and revised the legend for Figure 1.

Figure 7A and 7B: The authors might want to reverse x and y axis to match the y-axis of Figure C.

RESPONSE: While we appreciate this suggestion, we think the data are more appropriately presented in their current form. Since QR and C-peptide change are clinical outcomes, we have shown them as the dependent variables on the y-axis. Similarly for Figure C, we assess if age (independent variable) influences the biomarker shown on the y-axis.

Line 702: Is this generalized linear models or just logistic regression?

RESPONSE: We thank the reviewer for catching this detail. This analysis is based on logit-link binomial generalized linear models; we have updated lines 675 and 783 in the legend for Figure 8 with this information.

Line 727: Please clarify the weighting.

RESPONSE: We have added further details regarding the weighting into the manuscript. Weighting was incorporated to account for the imbalance between classes (placebo n=448, and active treatment n=259). Active treatment observations were weighted more heavily than placebo observations; weights were determined as the target proportion divided by the actual proportion, where the target proportion is equivalent to the proportion of the dataset composed of placebo participants (i.e., the larger group, 63%). This is now described in lines 699-706 and 808-813 in the legend for Figure 9.

Figure 9: Would an ROC-curve type analysis be beneficial here?

RESPONSE: We appreciate the reviewer's suggestion; however, in this instance, we do not believe that an ROC-curve type analysis would be beneficial. Figure 9 serves to demonstrate how the QR distribution can be utilized to better understand which QR values are more likely to be associated with treatment response, while also cautioning about the challenges in discriminating placebo individuals from actively-treated individuals. An ROC-curve type analysis would be appropriate in proposing and evaluating specific cutoffs/thresholds for deciphering treatment response which is not the aim of this analysis and figure. Here we provide data to illustrate that while pre-specified cutoffs/thresholds may be suggested for future trials, these should be chosen with the recognition that a particular QR cutoff will not necessarily distinguish individuals who received an efficacious therapy from placebo individuals due to the large overlap in QR values between active and treated individuals.

Reviewer #3 (Remarks to the Author)

The authors re-examine the results of Phase 2 trials aimed at preserving endogenous insulin secretion in individuals at high risk of progressing to type 1 diabetes suggesting a recently-described potential refinement of the accepted (2 hour C-peptide AUC) regulatory outcome measure (named "QR") which adjusts for baseline values and age making use of previous data on C-peptide trajectories - hence reporting the difference between the observed and predicted values. The authors demonstrate that in some trials random "positive" effects in the control group can be seen to mask potentially useful treatment effects when reviewed in the context of other control groups from similar trials. This approach may also offer a way of reducing sample size for Phase 2 proof of concept trials, better predicting "responders," and moving to platform type trials.

Comments

- 1) The arguments set out are overall convincing
- 2) Can the authors comment on the reason for quite large mean increases in C-peptide secretion in control groups of the studies with teplizumab and canakinumab?

RESPONSE: We thank the reviewer for this comment. In working to address this question, we realized that our original figure 6 did not present the data clearly. We acknowledge that the reviewer is correct that C-peptide uniformly declines over time; this data is presented in Figure 2. However, we are not showing C-peptide over time in figure 6. Instead, figure 6 was meant to represent the idea that QR may allow trialists to consider designing trials to evaluate the efficacy of therapy at 6 months rather than 12 months after randomization. For this analysis, we calculated the QR at each time point separately. By connecting the lines from each time point, the original figure implied changes over time. However, QR is the difference between predicted and observed C-peptide values at each time point – thus, a clearer presentation of the data was needed. We have now provided an alternate figure allowing for ready comparison of the QR result at 6 months compared to 12 months for each trial. (See Figure 6 and its legend for edits, as well as lines 189-195 of the results.)

- 3) Is there a possibility that this approach could be over-sensitive i.e. allow weaker candidate molecules to progress to Phase 3? It should be emphasised that the ultimate outcome measure is delaying the onset of clinical type 1 diabetes by a meaningful length of time.

RESPONSE: We agree with the reviewer that treatment of at-risk individuals with disease modifying therapy to delay the onset of clinical disease is an important goal. However, it is not the only goal; treatment of those with type 1 diabetes with the aim of preserving beta cell function is an important outcome (see edits, lines 57-59).

Making decisions about next steps in both situations (i.e. further trials in newly diagnosed, or moving to prevention studies) is challenging. As noted in our paper, there are underappreciated limitations of selecting candidates for further study in both cases due to both type 1 and type 2 errors from single relatively small randomized-controlled trials in those with clinical type 1 diabetes. There is the additional problem in moving to prevention due to the different populations involved. However, we take the reviewer's point (and those from another reviewer) that QR may also have limitations if utilized to identify candidate therapies. We now describe that QR may be somewhat insensitive if used as an early outcome– that is, we may miss promising therapies using QR as an early outcome. (lines 190-192).

- 4) The comments regarding using similar approaches in other conditions seem a little off topic and could be toned down - adjusting for baseline value is quite widely performed for other outcome measures.

RESPONSE: As suggested by the reviewer, we have removed some sentences about using this approach in other conditions (lines 87-88, 289-292, 309-310, and the final sentence of the manuscript.) We also modified the title to state that our work focuses on T1D.

- 5) The authors should clarify that their comments on a potential lack of requirement for contemporaneous controls are restricted to Phase 2 trials

RESPONSE: We agree with this comment and have revised the text considerably to clarify (for example, lines 304-307, 304, and 309-315). Please see also our response to Reviewer 1, point A, which identified a similar concern.

6) The individual who originally described the QR measure is a member of the same consortium - is there a reason why he or she is not included here in the author group?

RESPONSE: All TrialNet papers have undergone review and approval by TrialNet's publication committee including TrialNet biostatisticians. The work for this paper was conducted by the listed authors.

REVIEWERS' COMMENTS

Reviewer #1 (Remarks to the Author):

The authors addressed my comments well.

Reviewer #2 (Remarks to the Author):

The authors have responded well to the concerns raised and have added important clarifications.

Reviewer #3 (Remarks to the Author):

The manuscript is greatly improved.

However, I do not understand the response to my comment: "can the authors comment on the reason for quite large mean increases in C-peptide secretion in control groups of the studies with teplizumab and canakinumab?"

It is stated in the response that "our original Figure 6 did not present the data clearly" and "We have now provided an alternate figure" - but Figure 6 seems identical in the revision (in addition the text of the legend even though highlighted in yellow seems unchanged from the previous text)?

My comment was actually based on Figure 5 (although I now realise I meant alefacept and canakinumab).

Second Response to reviews, May 2023

Reviewer #1 (Remarks to the Author):

The authors addressed my comments well.

Reviewer #2 (Remarks to the Author):

The authors have responded well to the concerns raised and have added important clarifications.

Reviewer #3 (Remarks to the Author):

The manuscript is greatly improved.

1. First comment from Reviewer #3.

However, I do not understand the response to my comment: "can the authors comment on the reason for quite large mean increases in C-peptide secretion in control groups of the studies with teplizumab and canakinumab?" It is stated in the response that "our original Figure 6 did not present the data clearly" and "We have now provided an alternate figure" - but Figure 6 seems identical in the revision (in addition the text of the legend even though highlighted in yellow seems unchanged from the previous text)?

Response: We believe that this reviewer is unfortunately mistaken as revised figure 6 and the legend are significantly different from figure 6 and the legend in the original version. This may have been difficult to see in the tracked version of the revised manuscript. Below are the two versions for easy comparison.

ORIGINAL FIGURE 6 with LEGEND

Figure 6. Effect of therapy determined at 6 months after randomization.

Mean QR (\pm 95% confidence interval (CI)) at interval time points. At 6 months, all positive trials at 1 year had QR greater than 0 and negative trials below zero. Since the QR metric is based on a linear model, where baseline is predictive of 1 year C-peptide, QR at interim timepoints was computed by deriving the expected C-peptide values at specified timepoints from the original QR equation, and determining the difference between the expected and observed values.

CURRENT REVISED FIGURE 6 WITH LEGEND

Figure 6. Effect of therapy determined at 6 months after randomization.

Mean QR \pm 95% confidence interval (CI) at (A) 6 months and (B) 12 months. All treatment arms with mean QR \pm 95% CI above 0 at 6 months were also above 0 at 1 year. Since the QR metric is based on a linear model, where baseline is predictive of 1 year C-peptide, QR at interim timepoints was computed by deriving the expected C-peptide values at specified timepoints from the original QR equation, and determining the difference between the expected and observed values. At 6 months: teplizumab n=51, alefacept n=31, rituximab n=52, 2 dose GAD-Alum (Diamyd Ph2) n=35, abatacept n=67, low-dose ATG n=29, ATG/GCSF n=28, imatinib n=43, 4 dose GAD-Alum (Diamyd Ph3) n=109, 2 dose GAD-Alum (Diamyd Ph3) n=107, MMF/DZB n=39, MMF n=28, canakinumab n=47, tocilizumab n=87, 3 dose GAD-Alum (TrialNet) n=44, 2 dose GAD-Alum (TrialNet) n=46, high-dose ATG n=36. Reference Table 1 for n at 12 months.

2. Second comment from Reviewer #3.

My comment was actually based on Figure 5 (although I now realise I meant alefacept and canakinumab).

(Editorial note: The original review stated "can the authors comment on the reason for quite large mean increases in C-peptide secretion in control groups of the studies with teplizumab and canakinumab?")

Response: We thank the reviewer for clarifying the original question. Regarding the specific question about the alefacept and canakinumab trials, the C-peptide AUC data for the control arms is presented in figure 2. As shown, in both trials the mean (\pm SD) C-peptide does fall from baseline to 1 year (Alefacept 0.64 \pm 0.22 at baseline and falls to 0.53 \pm 0.39 at 1 year; Canakinumab 0.618 \pm 0.29 at baseline and falling to 0.403 \pm 0.29 at 1 year). In figure 5, the reviewer is observing the range in QR as an outcome in control groups, emphasizing that trials with small N's have a random probability of higher or lower than expected outcomes at 1 year. For example, while – as noted by the reviewer - in these two trials the point estimate exceeds the expected value, for the low dose ATG trial, the point estimate is lower than predicted.